# Carbon system state determines warming potential of emissions

**Alexander J. Winkler**[1,2]*, **Ranga Myneni**[3], **Christian Reimers**[1], **Markus Reichstein**[1], **Victor Brovkin**[4,5]

**1** Max-Planck-Institute for Biogeochemistry, Jena, Germany, **2** Guest at Max-Planck-Institute for Meteorology, Hamburg, Germany, **3** Department of Earth and Environment, Boston University, Boston, MA, United States of America, **4** Max-Planck-Institute for Meteorology, Hamburg, Germany, **5** Center for Earth System Research and Sustainability, University of Hamburg, Hamburg, Germany

* awinkler@bgc-jena.mpg.de

**Data Availability Statement:** CMIP5 and CMIP6 model output can be obtained at the ESGF nodes (https://esgf.llnl.gov/nodes.html). Processed model output and plotting data are available under https://zenodo.org/records/10905388. Because the

## Abstract

Current strategies to hold surface warming below a certain level, *e. g.*, 1.5 or 2°C, advocate limiting total anthropogenic cumulative carbon emissions to $\sim 0.9$ or $\sim 1.25$ Eg C ($10^{18}$ grams carbon), respectively. These allowable emission budgets are based on a near-linear relationship between cumulative emissions and warming identified in various modeling efforts. The IPCC assesses this near-linear relationship with high confidence in its Summary for Policymakers (§D1.1 and Figure SPM.10). Here we test this proportionality in specially designed simulations with a latest-generation Earth system model (ESM) that includes an interactive carbon cycle with updated terrestrial ecosystem processes, and a suite of CMIP simulations (ZecMIP, ScenarioMIP). We find that atmospheric $CO_2$ concentrations can differ by $\sim 100$ ppmv and surface warming by $\sim 0.31$°C (0.46°C over land) for the same cumulated emissions ($\approx 1.2$ Eg C, approximate carbon budget for 2°C target). $CO_2$ concentration and warming per 1 Eg of emitted carbon (Transient Climate Response to Cumulative Carbon Emissions; TCRE) depend not just on total emissions, but also on the timing of emissions, which heretofore have been mainly overlooked. A decomposition of TCRE reveals that oceanic heat uptake is compensating for some, but not all, of the pathway dependence induced by the carbon cycle response. The time dependency clearly arises due to lagged carbon sequestration processes in the oceans and specifically on land, *viz.*, ecological succession, land-cover, and demographic changes, *etc.*, which are still poorly represented in most ESMs. This implies a temporally evolving state of the carbon system, but one which surprisingly apportions carbon into land and ocean sinks in a manner that is independent of the emission pathway. Therefore, even though TCRE differs for different pathways with the same total emissions, it is roughly constant when related to the state of the carbon system, *i. e.*, the amount of carbon stored in surface sinks. While this study does not fundamentally invalidate the established TCRE concept, it does uncover additional uncertainties tied to the carbon system state. Thus, efforts to better understand this state dependency with observations and refined models are needed to accurately project the impact of future emissions.

amount of data is too large, raw model output from the individual emission pathway simulations using MPI-ESM1-2-LR is only available upon request from the corresponding author. Code and documentation for MPI-ESM version 1.2.01 (mpiesm-1.2.01-release, revision number 9234) for scientific users can be obtained at https://code.mpimet.mpg.de/projects/mpi-esm-license. Analysis and plotting scripts are available at public repositories (upon publication).

**Funding:** The study was funded by the Deutsche Forschungsgemeinschaft (DFG, German Research Foundation) under Germany's Excellence Strategy – EXC 2037 'CLICCS - Climate, Climatic Change, and Society' – Project Number: 390683824". MR, CR and AJW acknowledge support by the European Research Council (ERC) Synergy Grant "Understanding and Modelling the Earth System with Machine Learning (USMILE)" under the Horizon 2020 research and innovation 415 programme (Grant agreement No. 855187). RBM and AJW acknowledge funding by the Alexander von Humboldt Foundation. RBM also acknowledges funding from NASA Earth Science Division. The funders had no role in study design, data collection and analysis, decision to publish, or preparation of the manuscript.

**Competing interests:** The authors have declared that no competing interests exist.

## Introduction

The near-linear translation of cumulative emissions to surface warming implies TCRE as being independent of the timing of emissions and that the response of the carbon-climate system does not depend on the temporal trajectory of previous emissions [1–7]. However, this could likely be an artifact of quasi-exponential forcing of models lacking a realistic representation of terrestrial biospheric processes and full coupling in translation of carbon emissions to $CO_2$ concentration [8, 9] (emission- *versus* concentration-driven). In particular, accounting for a land carbon sink modulated by processes that respond to emitted carbon on different time scales and resultant changes in atmospheric residence time can imbue TCRE with pathway dependence. These shortcomings may introduce previously unconsidered uncertainties into TCRE and the resulting remaining carbon budgett [2, 10–14], potentially affecting efforts to achieve surface warming targets outlined in the Paris Agreement [15]. In the following, we detail these shortcomings and outline the ESM simulations designed to address them in this study.

### Shortcomings in the prevailing modeling paradigm

$CO_2$ emissions translate into a change in global temperature, *i. e.*, TCRE, primarily via three key processes: terrestrial and oceanic $CO_2$ uptake, the radiative forcing due to atmospheric $CO_2$, and ocean heat uptake [7, 16]. These processes are governed by complex nonlinear mechanisms, but they appear to balance each other, so that TCRE emerges to be a constant in the coupled climate-carbon cycle system as shown by many modeling exercises [7, 10, 16]. However, the TCRE proportionality and its underlying properties may only hold for the prevailing modeling paradigm and because of the simplifications made in most modeling exercises, as an analysis of simulations of CMIP5 generation Earth system models has already shown [9].

Firstly, the prevailing modeling paradigm, *e. g.*, in the Coupled Climate Carbon Cycle Model Intercomparison Project community (C4MIP) [17, 18], is to force the system with a prescribed atmospheric $CO_2$ timeline (concentration-driven runs) which suppresses the carbon cycle feedback to atmospheric $CO_2$ as opposed to emission-driven runs. This paradigm tenaciously persists as it is assumed that the feedback is linear and thus can be reconstructed from concentration-driven runs [13, 19, 20]. However, Boer *et al.* [21] already demonstrated that the complexity and nonlinearity of the system in fully-coupled emission-driven runs cannot be simplified to a linear behavior. Accordingly, a recent paper reasserts the need for emissions-driven climate projections in the upcoming CMIP7 [22].

Secondly, the ESM modeling community focuses on studying TCRE in forcing trajectories that follow a (quasi-)exponentially increasing $CO_2$ concentration (*e. g.*, 1pctCO2 and RCP8.5 forcing) [13, 18–20]. Many key features of TCRE, *e. g.*, the assumption of the linear carbon-cycle feedback, might only hold for this specific scenario. Raupach [8] has analyzed in detail the special case of exponential forcing of the climate carbon cycle and has shown that many aspects of TCRE are scenario specific.

Thirdly, another key shortcoming of TCRE research until today is the focus on atmosphere-ocean coupling and the use of simplified modelling approaches therein, such as analytical models [2] or so-called ESMs of intermediate complexity [EMICs; *e. g.*, in] [3, 5, 23–25]. While these methods have advanced our understanding by revealing potential non-linearities and state dependency in TCRE, *e.g.*, [26–28], the latest studies employing comprehensive ESMs in investigating pathway dependence in fully-coupled emission-driven simulations date back already a decade [12, 29, 30]. These ESM used only a rudimentary representation of the terrestrial biosphere, overlooking vegetation dynamics. Winkler *et al.* [9], however, suggested that those models from the last generation of ESMs (CMIP5 models) with a more comprehensive representation of the terrestrial biosphere reflect strong pathway and state dependency in

carbon uptake processes. The C4MIP community also explicitly acknowledges that the lack of accounting for dynamic vegetation in these ESMs could lead to overly conservative estimates of the climate-carbon cycle feedbacks [18].

All three shortcomings outlined above, taken together, may be the reason why the climate-carbon cycle feedback connected to vegetation dynamics, and the resulting state dependency of the climate-carbon cycle system has been overlooked to date.

## Earth system simulation of idealized pathways from emission to relaxation

It is illustrative to look at the buildup of atmospheric $CO_2$ concentration during the emission phase and its drawdown during the relaxation phase for different pathways of identical total emissions to understand the evolution of the state of the carbon system and associated transient warming. Therefore, we conduct multiple emission-driven experiments with a state-of-the-art Earth system model (CMIP6 version of MPI-ESM1–2-LR; Materials and methods) that has updated terrestrial processes [31] (land nitrogen cycle, soil hydrology, vegetation carbon dynamics, fires, *etc.*) and thus address all major shortcomings in previous modeling studies mentioned above. We use four emission pathways (Fig 1a): constant (labelled "C"), exponential ("E"), negative parabolic ("P") and linear decay to zero from a high initial value ("L"). The total emitted carbon in all cases is fixed (Fig 1b), 1,200 Pg ($10^{15}$ grams) C, which is approximately the total allowable carbon budget to hold surface warming to 2°C above the pre-industrial value [14] (details in Section Materials and methods). The duration of the emission phase is either 100 or 200 years starting from the pre-industrial equilibrium of the system. These two distinct time horizons facilitate separate examination of the temporal influence alongside the diverse emission trajectories. The 200-year duration is similar to the 2°C scenario (Figure SPM.10 in IPCC [14]). For the 200-year experiments, we also simulate how the system evolves for 300 years after emissions have ceased, the so-called relaxation phase. Although these pathways are idealized, designed deliberately for the purposes of this study, both the timescale and magnitude of forcing are comparable to historical emission rates [32] and directly relevant to current climate change. Importantly, atmospheric $CO_2$ is treated here as a time- and space-variant tracer with its value determined by the residual of the various sinks and sources in the Earth system.

## Results

### Pathway-dependent atmospheric $CO_2$ accumulation and surface warming during emission phase

The accumulation of $CO_2$ in the atmosphere during the emission phase differs across the four pathways even though the same amount of carbon is emitted in all cases (Fig 1c). Only those pathways with declining emissions ("P" & "L") show decreasing atmospheric $CO_2$ concentration during the emission phase. Changing the duration from 200 years to 100 results in a 40–60 ppmv difference (Fig 1c and S1 Fig). The concentration upon full emission is 100 ppmv lower if most of the forcing is applied in the first half of the emission phase (75% in "L" vs. 20% in "E" pathway; Fig 1c). This suggests a delayed response of processes responsible for atmospheric carbon removal. These differences in $CO_2$ concentration are consequential—they lead to ∼0.31°C difference in surface warming (0.46°C over land and 0.25°C over the oceans; Fig 2). The inter-pathway differential is bigger in the longer 200-yr runs (∼0.2°C) compared to the 100-yr runs (∼0.1°C), because processes operating on long timescales come into play in the longer simulations (Table 1). Thus, the TCRE differs between pathways and duration (Fig 1d) by as much as ∼0.26°C/(Eg C)—almost twice this for land (S2 Fig). Here TCRE is

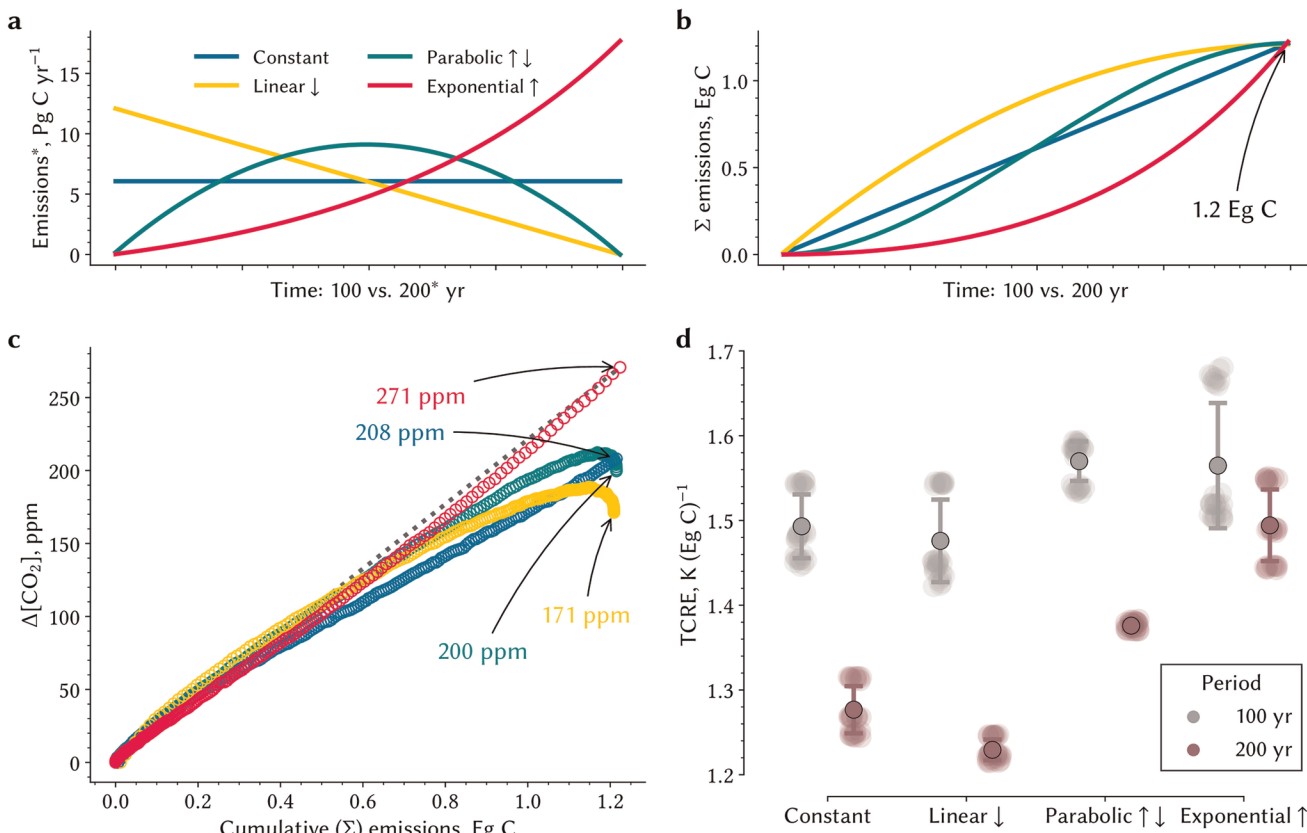

**Fig 1. Change in atmospheric CO₂ concentration, and thus surface warming, differs significantly across the different pathways for the same amount of emitted carbon. a** The colored lines depict the four different pathways with constant (blue), negative parabolic (green), linearly decreasing (yellow), and exponentially increasing carbon emissions (red). The pathways are either realized over a time period of 100 or 200 years (*x*-axis). The annual emission rates are given for the 200 year period as marked by the * (*y*-axis). **b** The colored lines depict the time integrals of the four pathways shown in a, where all accumulate to 1.2 Eg emitted carbon in the final year. **c** Relationship between cumulative carbon emissions (*x*-axis) and the change in atmospheric CO₂ concentration ($\Delta[CO_2]$) is shown for 200 years, where the colors refer to the four different pathways analogous to a. Each circle refers to one year and the arrows annotate the final $\Delta[CO_2]$ in each model experiment. See S1 Fig for a comparison of 100 vs 200 year runs. **d** Transient Climate Response to Cumulative Emissions (TCRE) for different pathways (*x*-axis) and simulation periods (colors). TCRE is estimated using the conventionally used linear regression method [13]. Shaded dots exhibit the spread in the estimates of the final five years when 1.2 Eg C have been emitted as well as among different realizations. Pointplot with whiskers show the mean and standard deviation of the spread. Companion S2 Fig shows Land TCRE.

estimated using a expanding-window regression approach (Materials and methods, [13]) based on the relationship of change in temperature *versus* cumulative emissions depicted in Fig 2. Notably, peak warming can occur before even reaching net-zero CO₂ emissions, as observed in the "L" experiment (Fig 2b). The intransigence of TCRE with respect to duration in the case of exponential forcing could be a specific anomaly, as discussed in [8], and not a general property of the system (Fig 1d), as previously suggested [20, 33–35]. These differences in atmospheric CO₂ concentration and TCRE illustrate a dependency of the carbon-climate system on the forcing sequence and pathway duration—note that cumulative emissions in all cases are identical, *viz.* 1,200 Pg C.

## Decomposition of TCRE in its individual responses within the coupled carbon-cycle climate system

To elucidate the TCRE, we dissect it into its key components: physical climate feedbacks, planetary heat uptake, radiative forcing, and carbon cycling (Materials and methods, [13]). First,

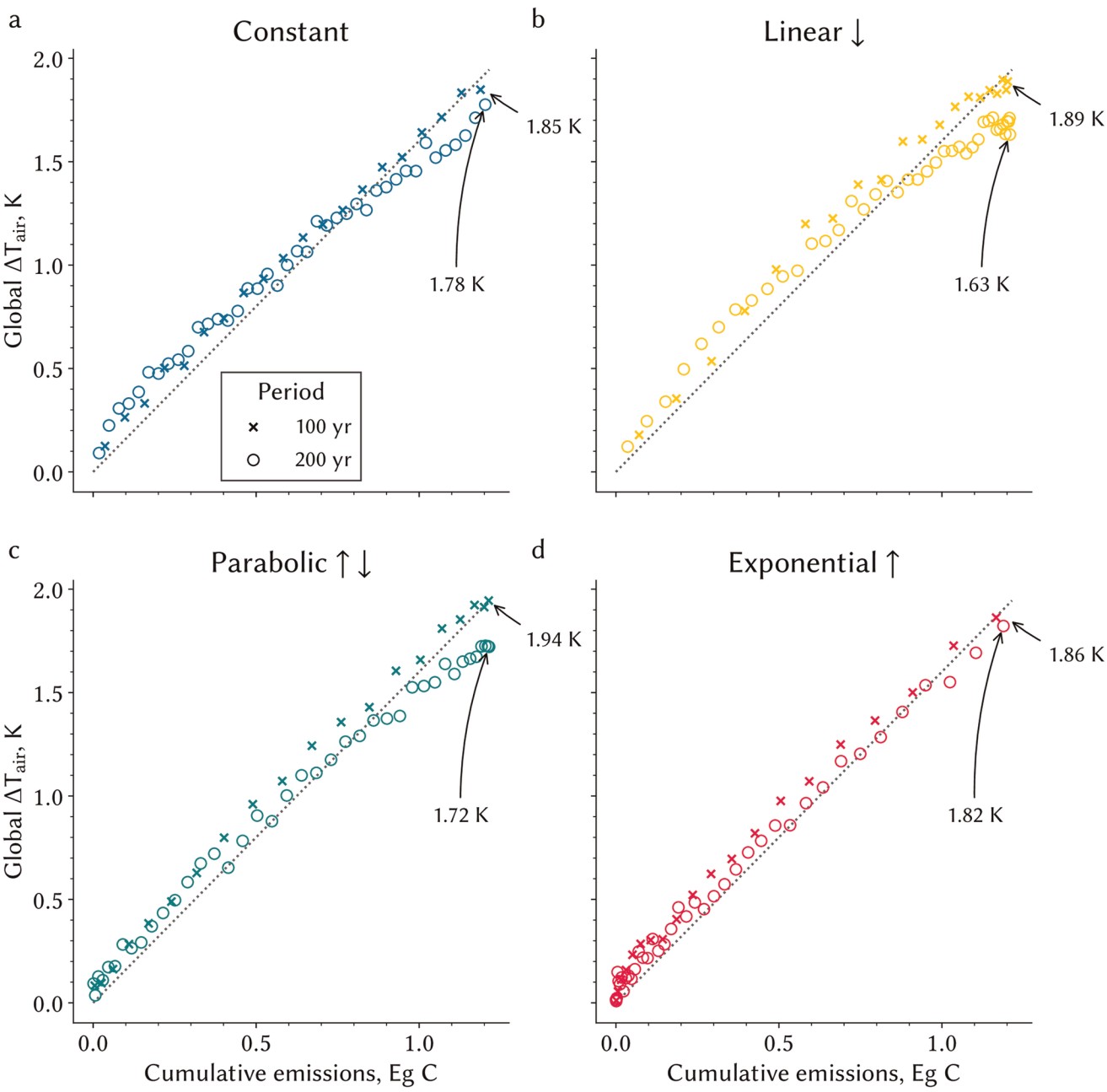

**Fig 2. Change in global mean surface air temperature across the different pathways and different periods for the same amount of emitted carbon. a–d** Relationship between cumulative carbon emissions (*x*-axis) and the change in global mean air temperature ($\Delta T_{air}$) is shown, where the colors refer to the four different pathways analogous to Fig 1 and the markers refer to the different time periods. To reduce inter-annual variability, five-year averages across the three realizations are displayed. The arrows annotate the final $\Delta T_{air}$ in each model experiment. The dotted black line facilitates comparison between the pathways.

TCRE is broken down into the Transient Climate Response (TCR), representing surface temperature change relative to atmospheric carbon change, and the airborne fraction (AF), expressed in cumulative terms, quantifying atmospheric carbon change against cumulative emissions (Eq 6). In accordance with Fig 1c, the AF reflects a strong pathway dependence induced by the carbon cycle response (Fig 3b). For instance, in the "L" experiment, only 30%

**Table 1. Change in annual mean surface air temperature at the end of the emission phase compared to pre-industrial conditions, aggregated globally and separated for land and ocean.**

| Pathways | Period | Constant | Linear ↓ | Parabolic ↑↓ | Exponential ↑ |
|---|---|---|---|---|---|
| *Unit* | yr | K | K | K | K |
| Global $\Delta T_{Air}$ | 100 | 1.85 ± 0.07 | 1.89 ± 0.03 | 1.94 ± 0.15 | 1.86 ± 0.11 |
| | 200 | 1.78 ± 0.03 | 1.63 ± 0.05 | 1.72 ± 0.072 | 1.82 ± 0.08 |
| Ocean $\Delta T_{Air}$ | 100 | 1.53 ± 0.05 | 1.59 ± 0.02 | 1.62 ± 0.11 | 1.56 ± 0.10 |
| | 200 | 1.49 ± 0.02 | 1.37 ± 0.06 | 1.45 ± 0.05 | 1.52 ± 0.08 |
| Land $\Delta T_{Air}$ | 100 | 2.64 ± 0.13 | 2.62 ± 0.05 | 2.74 ± 0.26 | 2.61 ± 0.15 |
| | 200 | 2.49 ± 0.06 | 2.28 ± 0.08 | 2.39 ± 0.11 | 2.56 ± 0.09 |

Displayed values reflect the average over the last five years and three realizations for the four different pathways over 100 and 200 years. Uncertainty is indicated by one standard deviation. See Fig 2 for temperature change evolution with increasing emissions in 100 and 200 year runs.

of emissions remain airborne compared to 47% in the "E" experiment. The TCR buffers for some pathway dependence induced by the carbon cycle response; hence, the "E" experiment exhibits the least warming and "L" the most for the same atmospheric carbon burden (Fig 3c). Next, TCR is decomposed into the constituents radiative forcing for given atmospheric carbon change and the subsequent thermal response (Eq 7). The logarithmic relationship of the radiative forcing to atmospheric $CO_2$ explains only some of the pathway dependence (Eq 9; Fig 3d), because the thermal response retains most of the TCR signal (compare Fig 3c & 3e). Finally, the thermal response is factorized into the radiative forcing that heats the surface, *i.e.*, the fraction not absorbed by oceans, and the inverse physical feedback parameter $\lambda^{-1}$, representing the system's radiative response to surface warming (Eq 8). The λ parameter encompasses the radiative response components such as the Planck feedback, lapse rate, relative humidity, surface albedo, and cloud feedback [13]. Only small deviations in $\lambda^{-1}$ across pathways confirms the expectation that underlying atmospheric processes respond to forcing on short time-scales ([36], Fig 3g). However, an analysis of the apportionment of radiative forcing into ocean warming and surface warming points out notable differences among the pathways (Fig 3f). While the oceans in the "E" experiment continue to absorb the majority of the trapped energy in the final decades of the simulation, the ocean heat absorption in the "L" experiment already ranges at a higher saturation level, thereby allocating more energy for surface warming.

The above follows the rationale of several studies, including [3, 10, 13, 37], which demonstrated that the balance of oceanic heat and carbon uptake moderates pathway dependence in TCRE. Our experiments, however, reveal a nuanced understanding of this relationship. We find that considering an improved representation of the terrestrial biosphere, reflecting various time-scales, introduces non-negligible pathway dependence into TCRE. In other words, ceanic heat uptake compensates for some, but not all, of the pathway dependence induced by the carbon cycle response (Fig 3b and 3g). Only small contributions to this compensating effect can be attributed to the saturation of radiative forcing with increasing atmospheric $CO_2$ levels (Fig 3f). Now, we focus on a closer examination of the response and impact of the terrestrial biosphere and its diverse timescales within the coupled system.

## The lagged response of the terrestrial biosphere in anthropogenic carbon sequestration

The sequestration of anthropogenic carbon in the atmosphere into the sinks involves several inter-connected processes operating at different timescales [7, 38]. The timescales of oceanic

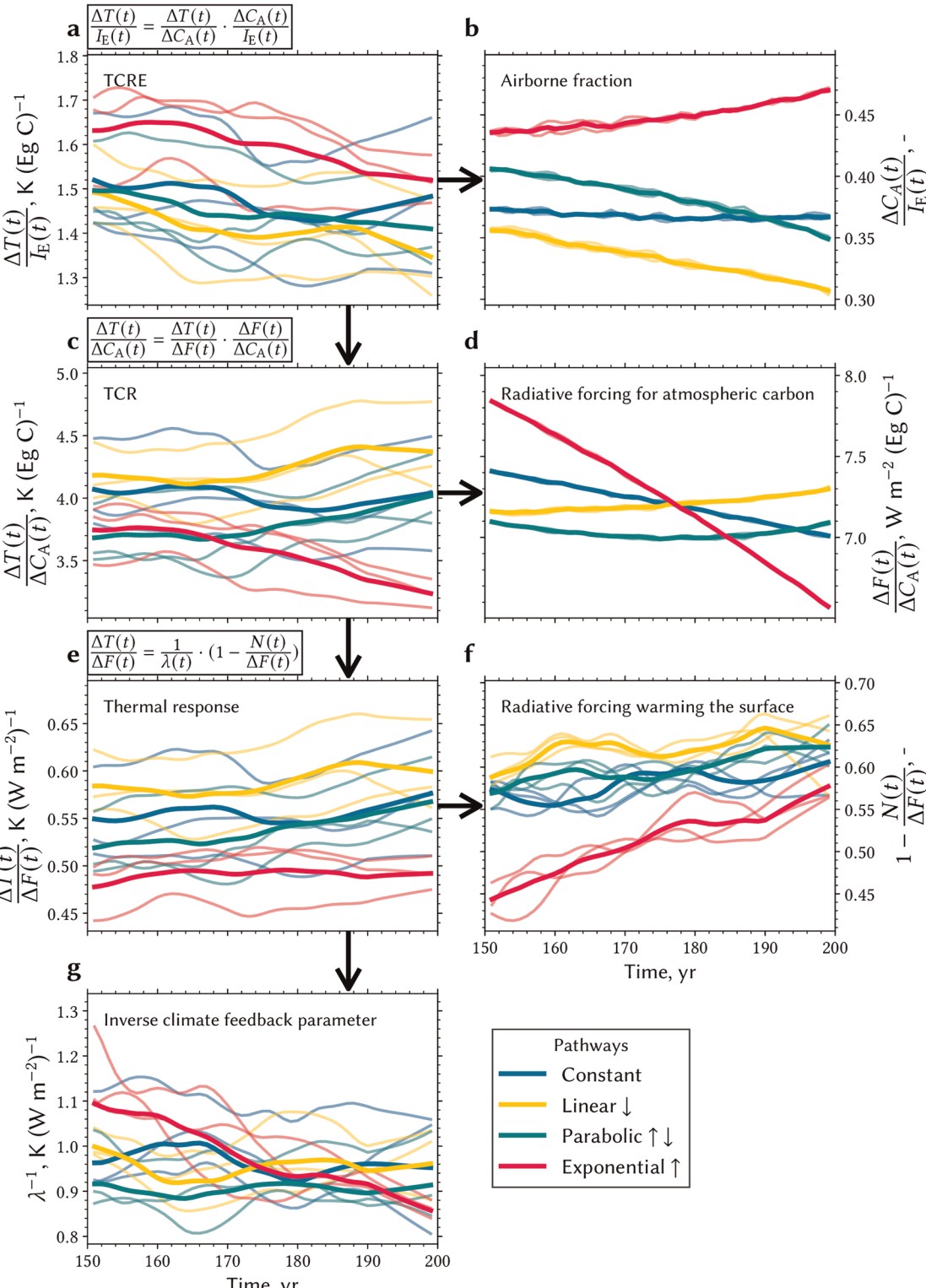

**Fig 3. Decomposition of TCRE into planetary heat uptake, carbon cycling, and physical feedback parameter. a** The evolution of TCRE in the four different emission pathways over the last five decades in the 200-year simulations. Each thick line represents the ensemble average of three realizations per experiment smoothed by a low-pass filter (Savitzky-Golay filter). Following the black arrows, TCRE is first decomposed into the airborne fraction of emissions (expressed in cumulative terms, *i. e.*, atmospheric carbon change over cumulative emissions; **b**) and the Transient Climate Response (TCR, **c**). TCR is further decomposed into the radiative

forcing of atmospheric carbon (**d**) and the thermal response (**e**), which is again decomposed into the fraction of radiative forcing warming the surface (**f**) and the inverse climate feedback parameter (**g**). The equations displayed at the top of **a**, **c** and **e** describe the three steps of the TCRE factorization and are explained in more detail in the Eqs 5 to 9 in Materials and Methods.

uptake are determined by processes that generally saturate with cumulative emissions, and have been well studied [3, 39, 40]. Carbon is first absorbed into the mixed layer of the surface ocean within years to decades. The carbon is then mixed through ventilation of the upper ocean over decades to centuries. Transport to abyss by circulation occurs on slower timescales of centuries, or even millennia.

The timescales of carbon sequestration are less clear in the case of terrestrial biosphere where lagged processes tied to ecological succession, land-cover, and demographic vegetation dynamics come into play [41, 42]. Our simulations with the CMIP6 version of the MPI-ESM1–2-LR, one of the few models that resolves these processes [18], show pathway dependent increases in tree cover and associated carbon pool (Fig 4a and 4b and S3a and S3b Fig) commensurate with the forcing timeline. Slow processes of ecological succession and land-cover change in response to shifting bioclimatic boundaries are mainly realized during the emission phase when the forcing is applied earlier ("L" scenario) versus the relaxation phase ("E" scenario), *i. e.*, 300 years following cessation of emissions. The difference in forest expansion is most pronounced at high latitudes and in semi-arid regions—climatic zones where ecosystems respond strongly to climatic changes, which is in line with patterns of vegetation change seen in satellite observations [43]. Globally, this difference can be as large as $\sim 4 \times 10^6$ km$^2$ in tree cover ($\sim 100$ Pg C of carbon pool) at the end of the emission phase (Fig 4a and 4b). The maximal change in tree cover, however, tends to an asymptote in all scenarios given sufficient time in the relaxation phase (Fig 4c and S3c Fig). This asymptotic change is equivalent to about 20% increase of the pre-industrial cover.

## Pathway-dependent spillover effects into relaxation of atmospheric carbon burden

To better understand the delayed response of the global carbon cycle and, in particular, the role of the terrestrial ecosystems in it, we analyze a series of simulations of multiple ESMs, in addition to those described above. To this end, we first examine the effects of different emission pathways on atmospheric $CO_2$ drawdown during the relaxation phase, relying on simulations from the ZecMIP ensemble (Zero Emissions Commitment Model Intercomparison Project [44]; Materials and methods). The drawdown profiles are model- and pathway-specific (Fig 5a and 5b). The variation in atmospheric carbon at the end of the emission phase is larger across pathways than models (200 vs. 100 Pg C), which is contrary to expectation. All ZecMIP models, which are forced exponentially, show a higher atmospheric burden for a lower total amount of emitted carbon (1,000 Pg C) than the "C", "P" and "L" scenarios (1,200 Pg C). The drawdown in the exponential case is steepest due to relatively late forcing (80% during the second half of the emission phase) suggesting spillover effects into the relaxation phase of lagged processes responsible for carbon removal. The opposite is the case for the "L" pathway where much of the forcing is applied earlier in the timeline (75% during the first half of the emission phase). Notably, global cooling closely follows the trajectory of falling atmospheric $CO_2$ in each pathway (S4a and S4b Fig). Even a century after emission cessation, global mean surface temperatures remain divergent by up to 0.2 K, gradually converging as atmospheric $CO_2$ levels continue to fall.

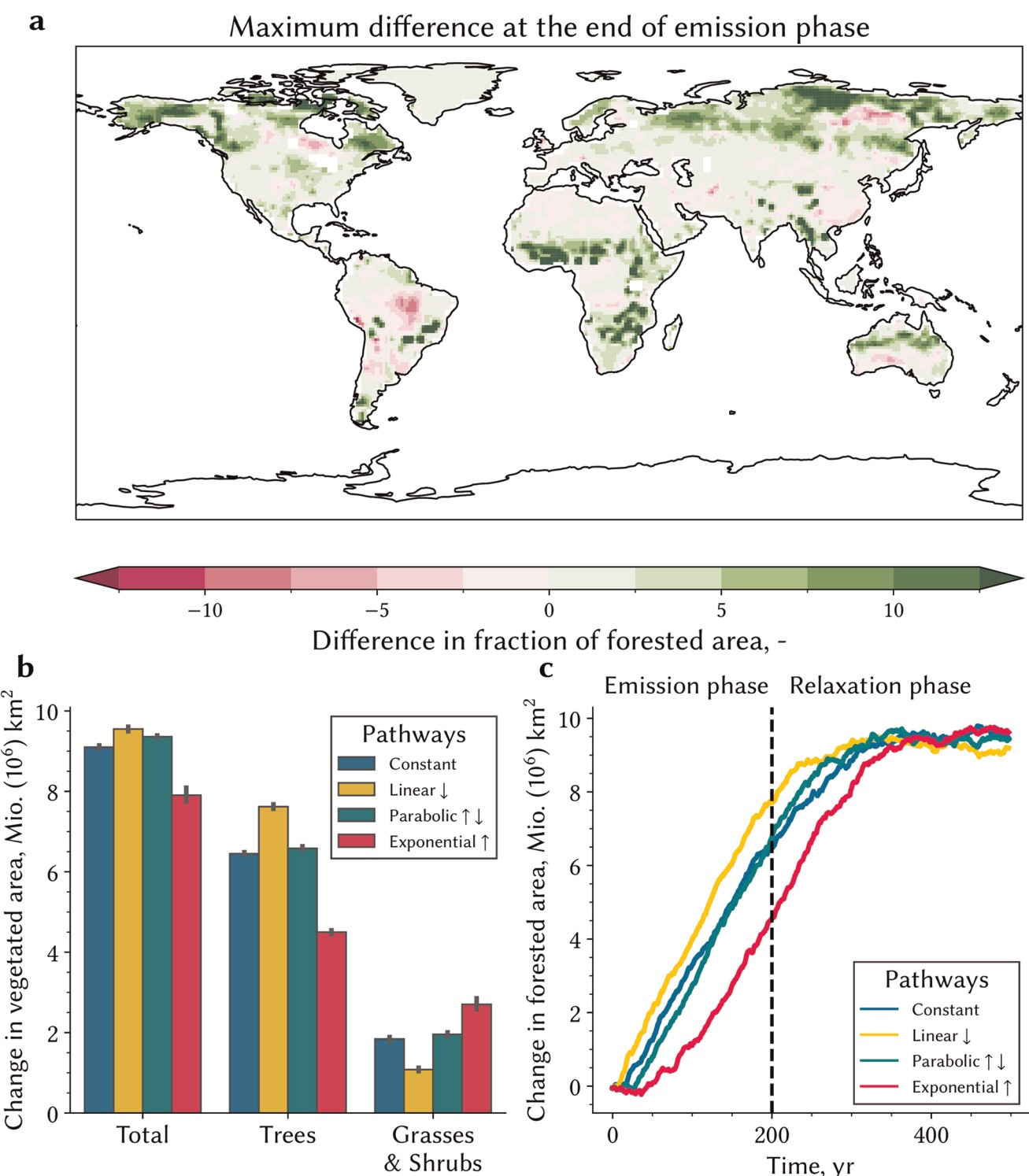

**Fig 4. Vegetation dynamics explain the pathway dependence in the terrestrial carbon cycle. a** Global map of the difference in the fraction of forested area between the L and E pathways (yellow and red in b and c, respectively) at the end of the emission phase, calculated with MPI-ESM1–2-LR in the 200-year runs. **b** Change in vegetated area for forest, grass- and shrublands, and total vegetation for the different emission pathways. The whiskers represent the uncertainty between the different realizations. **c** Change of forested area across the four pathways as a function of time for both the emission and relaxation phases. Companion S3 Fig shows changes in associated carbon pools.

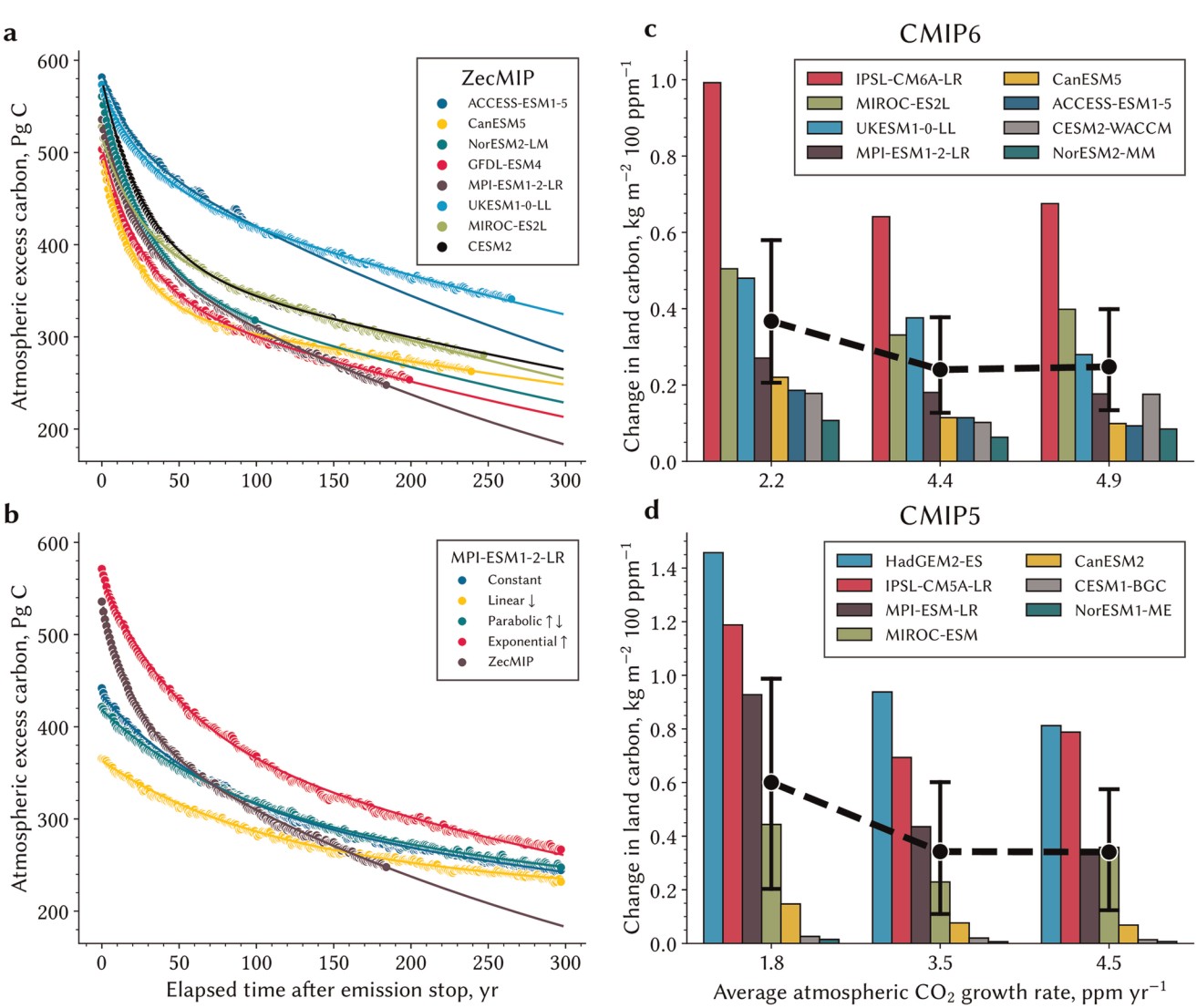

**Fig 5. Evolution of atmospheric excess carbon after emission stop and land carbon sensitivity to different CO₂ growth rates in CMIP ensembles. a** Different colored dots depict different CMIP6 models in ZecMIP, where the lines refer to the best fit using a double exponential decay function (see Materials and methods). **b** as in a but the colors refer to the different emission pathways and the ZecMIP run of the MPI-ESM1–2-LR model. **c** Change in the land carbon pool at latitudes above 60°N per a 100 ppm change in atmospheric CO₂ across three different future scenarios in CMIP6 reflecting different CO₂ growth rates (*x*-axis; see Materials and methods). The black dot refers to the multi-model mean value for each growth rate, where the whiskers represent the standard deviation. **d** as in c but for CMIP5. Companion S5 Fig shows changes in green leaf area index in CMIP6 and CMIP5 simulations.

This spillover effect is demonstrated by fitting an impulse-response function to the decay of atmospheric carbon [45, 46] (Materials and methods) S1 Table). A fit using two exponentials, conceptually corresponding to two sink compartments, describes quite well the drawdown process across the four pathways and all ZecMIP models (Fig 5a and 5b). The two decay rates, $\tau_{slow}$ and $\tau_{fast}$, reflect the respective *e*-folding times—the timescale over which the carbon content of the compartment decreases to $1/e$ of its initial value ($\sim 0.37$). All ZecMIP models, which are forced with the same pathway, agree on the timescales and cluster around $750 \pm 250$ yr for $\tau_{slow}$ and around $25 \pm 4$ yr for $\tau_{fast}$. The decay rates, however, vary across the pathways for the same model MPI-ESM1–2-LR (S2 Table, Fig 5b). For example, the slow (fast) timescale is around 3500 yr (100 yr) in the "L" pathway (most forcing early in the emission phase) and is

only around 750 yr (50 yr) in the "E" pathway (most forcing later in the emission phase). We employ a sampling of various parameter combinations to assess uncertainty when fitting timescales of millennia to decay time series that only span centuries (S2 Table, Materials and methods). It is important to note that the reported decay timescales do not claim correctness but rather underscore the variation in atmospheric carbon decay among pathways, surpassing the bounds of uncertainty. Thus, the emergent decay rates are not a model-specific property but are contingent upon the forcing-induced state of the system, contrary to earlier reporting [45], yet consistent with efforts to account for state dependency [26].

### Carbon-climate system dependency on the forcing timeline as indicated in other Earth system models

The profiles of atmospheric $CO_2$ concentration during the emission and relaxation phases depict a carbon-climate system dependency on the forcing timeline and pathway duration that is largely tied to delayed processes in the terrestrial biosphere (Fig 4). This dependency can also be gleaned from earlier CMIP5 and recent CMIP6 simulations. But first, a couple of caveats are in order. These model intercomparison exercises lack dynamics arising from an interactive carbon cycle as these are $CO_2$ concentration driven simulations only [47, 48]. And, only a few models in these ensembles include vegetation dynamics explicitly, *e. g.*, inter-species competition from changing bioclimatic constraints and nutrient resources. Nevertheless, it is possible to distill the response of natural vegetation, *i. e.*, vegetation north of 60˚N, for a fixed increase in $CO_2$ concentration (Materials and methods, [9]). Most models show a stronger increase in green leaf area and higher land carbon storage for scenarios (RCPs or SSPs) with lower atmospheric $CO_2$ growth rate, *viz.* cases where the same forcing (*e. g.*, 100 ppmv) is applied over a longer period (Fig 5c and 5d and S5a and S5b Fig)—approximately 50–75% more carbon is stored in land sinks in low-growth rate scenarios than in high-growth rate scenarios. This time dependency in the terrestrial carbon cycle, hitherto overlooked, serves to highlight the dependency of the system on forcing timeline.

### Concomitant land and ocean $CO_2$ uptake is independent of the emission pathway

The time dependency in the carbon system arises principally due to lagged carbon sequestration processes in the oceans and on land. This dependency imbues a state to the carbon system, and because of bidirectional linkages, to the climate system as well. The temporally evolving state of the carbon system can be characterized by the cumulative carbon sink. This definition embodies the requisite memory property of the system by integrating historical sink fluxes to a pool of anthropogenic carbon, and thus responses to current and future emissions depend on the trajectory of past emissions. The definition is also unambiguous because the sink shares of land versus ocean emerge to be pathway independent, *i. e.*, the ratio of land to ocean sink shares follows the same nonlinear relationship irrespective of pathway properties, *i. e.*, emissions timeline, duration, and cumulative amount (Fig 6a and 6b). The land-ocean relationship based on historical sink amounts derived from the Global Carbon Project, which synthesizes observation-based and multi-model estimates, closely tracks the relationship identified by MPI-ESM1.2-LR. In general, the land sink dominates the transient response and thus the time dependency of the system during the emission phase (Fig 6a), in line with most CMIP models [18]. Oceanic uptake begins to dominate in the relaxation phase as the land sink saturates (Fig 6b). The pathway invariance breaks down when the land becomes a source suggesting a fundamental change in the carbon system, centuries after cessation of emissions. Additional simulations with an Earth System Model of Intermediate Complexity (EMIC) CLIMBER-2

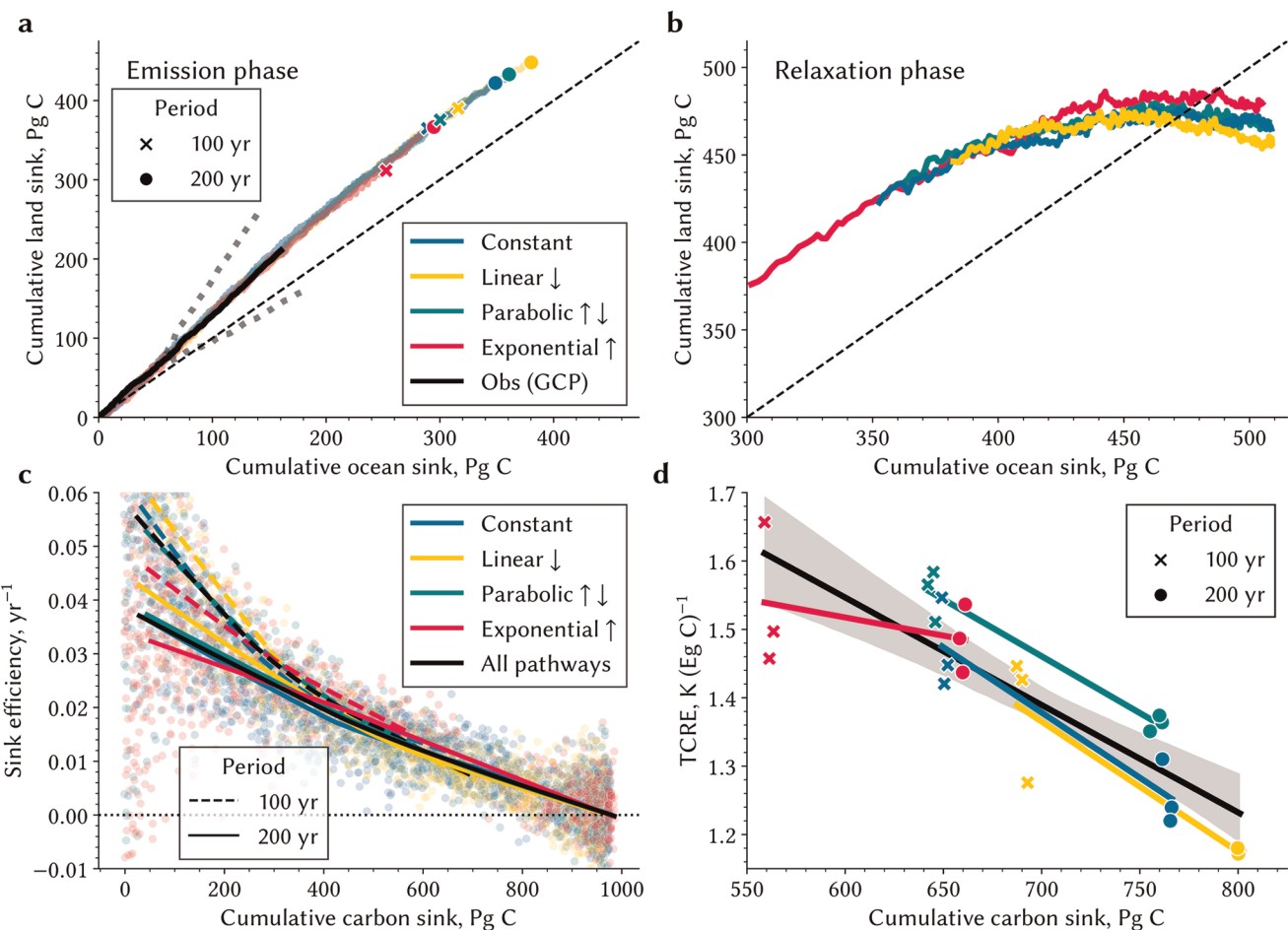

**Fig 6. State of the carbon system, sink efficiency, and TCRE across the pathways. a** Cumulative land and ocean sinks are juxtaposed across all pathways and both simulation periods. The markers denote the final state at the end of the emission phase of 1.2 Eg C. The dashed line indicates equal magnitude of cumulative land and ocean sinks. The black line refers to the average relationship inferred from GCP estimates, where the gray dots follow plus/minus the joint uncertainty for both cumulative land and ocean sink. **b** As in a, but for the relaxation phase. **c** Sink efficiency is plotted as function of the cumulative carbon sink for both emission and relaxation phase. The lines represent a nonparametric LOWESS fit, with the solid (dashed) line referring to the 200-(100-)year simulations in the different pathways indicated by different colors. The black lines indicate the relationship when all pathways are fitted at once. **d** TCRE is plotted as function of the cumulative carbon sink, *i. e.*, the state of the carbon system. The colored lines represent a linear fit for the four different pathways. The black lines indicate the relationship when all pathways and both periods are fitted linearly at once. The grey shading reflects uncertainty based on an 95% confidence interval estimated using bootstrapping. Companion S7 Fig shows the land-ocean sink relationship during the emission and relaxation phase for the EMIC CLIMBER-2.

[49] corroborates the described behavior and the emergent pathway invariance during the emission phase (S7a and S7b Fig). These lines of evidence support the robustness of the pathway-invariance of land-to-ocean apportionment of sequestered anthropogenic carbon. However, this behavior of the carbon system must be further tested to better understand the underlying dynamics and implications.

## Carbon system state determines sink efficiency and warming potential of emissions

The state of the carbon system is uniquely related to sink efficiency and TCRE (Fig 6c and 6d) in a manner none of the aforementioned properties of emission pathway are. Here, sink

efficiency $k_S$ is defined as the combined flux into the land ($f_L$) and ocean ($f_O$) for a given level of atmospheric carbon ($C_A$) loading [$k_S = \frac{-f_L - f_O}{C_A}$ [8]; (Pg C yr$^{-1}$) (Pg C)$^{-1}$]. The uptake rate into sinks, normalized for atmospheric carbon load, depends on how much of anthropogenic carbon has already been sequestered, *i. e.*, the state, and not on any particular property of the emission pathway. For example, cumulative emissions of 1,200 Pg C via the four pathways ("C", "P", "L" and "E" in Fig 1a) over a certain duration (100 or 200 years) result in very different sink efficiencies (S6 Fig). That is because the cumulative sink amount, the state, varies substantially. However, at the same state of the system, the sink efficiencies converge among the pathways (Fig 6c).

Transient surface warming in response to $CO_2$ emissions also exhibits a similar relationship to the state (Fig 6d), not surprisingly, given the latter's relation to sink efficiency. Accordingly, 1,000 Pg of carbon emitted can result in different TCRE depending on the timing of emissions (Fig 1d), however, TCRE will be approximately constant for the same amount of carbon stored in sinks. For example, emissions of 1,200 Pg C in the "E" pathway over 200 years or in the "C" pathway over 100 years result in approximately the same state of the carbon system and hence similar TCRE. Other states in the physical climate system also evolve differently depending on the emission pathway, such as processes controlling ocean heat uptake as also evidenced by the TCRE decomposition analysis presented earlier (Fig 3; [3]). The variability in these processes may explain the smaller deviations in the relationship in Fig 6d; however, the carbon cycle state appears to dominate the TCRE pathway dependence.

## Conclusions

We conclude that the state of the carbon system is central to understanding and solving problems associated with anthropogenic injection of carbon into the atmosphere. It is the state, not any particular property of the emission pathway, that determines how much surface warming can be expected for given emissions. In simpler words, it is not the amount emitted, but the emitted-amount stored in sinks that determines the warming potential of emissions. Our model experiments suggest that the effect of the state becomes apparent when considering both a more comprehensive representation of the terrestrial and oceanic carbon cycle in a fully coupled carbon-climate system. In this setting, the processes that translate $CO_2$ emissions into a change in global temperature no longer balance out, so that TCRE emerges as a constant. Peak warming can even occur before reaching net-zero $CO_2$ emissions, prompting new considerations in the IPCC's discourse on when to expect surface warming to stabilize [50, 51].

While these non-linearities do not invalidate TCRE, they do introduce additional uncertainty. If the model-specific range of 0.25°C for TCRE, as quantified in this study, were to apply in all models in this fully-coupled setting, it could substantially widen the uncertainty range of TCRE. For example, it might increase the IPCC's "likely" range of uncertainty from 1.3°C [14] to 1.55°C, representing a 20% increase in uncertainty that linearly propagates into carbon budget calculations. Given this heightened uncertainty and TCRE's sole focus on surface warming, questions arise regarding its practical utility in shaping climate policy. Carbon emissions not only trigger surface warming but also affect various other aspects of the Earth system with varying impacts across different emission pathways. For example, earlier emissions might result in less surface warming, reduced ocean acidification, and sea-ice loss but could lead to higher thermosteric sea-level rise compared to a later timing of emissions (S8 Fig). Thus, future research must explore this multi-faceted pathway dependence in the Earth system and inform carbon budgets and emission trajectories to minimize the overall impact of emissions.

This study has shown that efforts must prioritize characterizing the current state of the carbon system using observations and refined models, such that the impact of future emissions can be more accurately projected. Critical to this endeavor must be a focus on vegetation dynamics and lagged responses in the terrestrial carbon cycle, as the number of models explicitly accounting for these processes has decreased in CMIP6, although the total number of participating models has increased [18].

## Materials and methods

### Max-Planck-Institute Earth system model

MPI-ESM1.2 is the latest version of the state-of-the-art Max Planck Institute Earth System Model, which participates in the sixth phase of the Coupled Model Intercomparison Project (CMIP6) [52]. Mauritsen *et al.* [31] describe in detail the model developments and advancements with respect to its predecessor, the CMIP5 version [53]. Here, we use the low resolution (LR) fully coupled carbon/climate configuration (MPI-ESM1.2-LR), which consists of the atmospheric component ECHAM6.3 with 47 vertical levels and a horizontal resolution of $\approx$ 200 km grid spacing (spectral truncation at T63). The ocean dynamical model MPIOM is set up on a bi-polar grid with an approximate grid-spacing of 150 km (GR1.5) and 40 vertical levels. MPI- ESM1.2-LR includes the latest versions of the land and ocean carbon cycle modules, comprising the ocean biogeochemistry model HAMOCC6 and the land surface scheme JSBACH3.2 [54].

As opposed to the high-resolution configuration, the LR variant of the MPI-ESM includes all the important processes relevant for longer time-scale changes of the land surface, such as a thoroughly equilibrated global carbon cycle, dynamical vegetation changes, interactive nitrogen cycle, a process-based fire model (SPITFIRE), and an interactive coupling of all sub-models. Crucially for this study, only the LR variant allows us to simulate a full coupling of the climate-carbon cycle system, *i. e.*, to treat $CO_2$ as a tracer in the system that is exchanged between the atmosphere, ocean, and land. Specifically, we used the initial CMIP6 release of the MPI-ESM version 1.2.01 (mpiesm-1.2.01-release, revision number 9234). The final CMIP6 release version includes further bug fixes, which only negligible influence long-term sensitivities of simulated land surface processes.

### Idealized pathway experiments

We design four idealized pathways that all emit a total of 1.2 Eg C following very different emission trajectories. We choose the amount of 1.2 Eg C as it approximates the reported "allowable" total carbon budget to meet the 2 K warming target ($\approx$1233 Pg C; IPCC AR5 [14]). In the four pathways, the budget is emitted at a constant emission rate ("C"), at an exponentially increasing rate ("E"), at a negative parabolic rate ("P"), and at an initially high emission rate that linearly decays to zero in the final year ("L") starting from the pre-industrial steady state (Fig 1a). The time integrals of all pathways, *i. e.*, the cumulative emissions at the end of the simulation, are identical (Fig 1b). We set the simulation time horizons to 100 as well as 200 years, where the latter is similar to the time horizon of the business-as-usual scenario for the 2 K carbon budget (compare Figure SPM.10 in the IPCC AR5 [14]). Although the pathways are idealized, both the time scales and the magnitude of forcing (the maximum emission rate approximates maximum observed emission rate) are within the range of the observed system [32]. In detail, the four pathways are constructed as follows:

**Pathway I: Constant emissions.** The first pathway assumes constant emissions over the entire time period. The equations for emissions $E_1(t)$ for each year $t$ can be represented as:

$$E_1(t) = \frac{K}{n} \tag{1}$$

where $K$ is the total carbon budget, and $n$ is the total number of years in the simulation period.

**Pathway II: Linear decreasing emissions.** In this pathway, emissions decrease linearly over time. The equation for emissions $E_2(t)$ for each year $t$ is given by:

$$E_2(t) = a_1 t + b \tag{2}$$

where $a_1$ and $b$ are coefficients determined such that the entire carbon budget $K$ is emitted, while the emission rate decreases linearly to zero at the end of the time period. These coefficients are calculated as follows:

$$b = \frac{2K}{n} \quad \text{and} \quad a_1 = -\frac{b}{n}$$

**Pathway III: Negative parabolic emissions.** In this pathway, the emission rate follows a negative parabolic trajectory over time. The equation for emissions $E_3(t)$ for each year $t$ is given by:

$$E_3(t) = a_2 t^2 + bt \tag{3}$$

where $a_2$ and $b$ are coefficients determined such that emissions start at zero, increase in a parabolic fashion, and reach full emission of $K$ at the end of the time period. These coefficients are calculated as follows:

$$b = \frac{6K}{n^2} \quad \text{and} \quad a_2 = -\frac{b}{n}$$

**Pathway IV: Exponentially increasing emissions.** In this emission pathway, the emission rate grows exponentially such that the entire carbon budget $K$ is emitted by the end of the time period. The emissions $E_4(t)$ for each year $t$ can be expressed as:

$$E_4(t) = a_3 \cdot (e^{rt} - 1) \tag{4}$$

where $a_3$ is a coefficient determined to achieve the desired total emissions, $r$ is the annual growth rate (in this case, 1% or 0.01), and $e$ is the base of the natural logarithm. The coefficient $a_3$ is calculated as:

$$a_3 = \frac{K}{(e^{rn} - 1)/r - n}$$

The $CO_2$ emissions in each pathway are uniformly distributed across the global land surface. However, high latitudes ($> 65°N/S$) and high mountain ranges ($> 2000m$) must be excluded to avoid numerical instabilities in the transport of carbon fluxes. These instabilities occur during stable atmospheric conditions in the boundary layer at night and/or winter, preventing mixing of emitted $CO_2$ with higher layers of the atmosphere, resulting in strong vertical gradients in the mixing ratio of $CO_2$, causing the model to crash. Atmospheric $CO_2$ is continuously derived as the net result of prescribed emissions and land-atmosphere as well as ocean-atmosphere fluxes as calculated by the respective submodel of MPI-ESM1.2-LR. To take the effect of climate

variability into account, we conduct three realizations for each pathway and time period by choosing different initial conditions from the CMIP6 pre-industrial control run.

## Zero Emissions Commitment Model Intercomparison Project (ZecMIP) in CMIP6

As a late addition to CMIP6 [52], the Zero Emissions Commitment Model Intercomparison Project (ZecMIP) was designed to quantify the amount of unrealized warming after emission cessation as a lagged response due to $CO_2$ already emitted to the atmosphere [44]. The physical and biogeochemical mechanisms underlying this delayed and therefore already "committed" climate response, the so-called "zero emissions commitment," are highly uncertain, but critical for considering whether the remaining carbon budget needs to be adjusted for the unrealized warming [6, 44].

Here we make use of the central and high-priority simulation in the ZecMIP ensemble termed *esm-1pct-brch-1000PgC*. The *esm-1pct-brch-1000PgC* simulation is a zero-emissions experiment with fully interactive $CO_2$, branched from the exponential pathway of 1% $CO_2$ annual increase (*esm-1pctCO2*) at the point of 1000 Pg C cumulative $CO_2$ emissions [44]. Because the compatible carbon emissions are reconstructed from the concentration-driven run *1pctCO2*, the point at which 1000 Pg C of cumulative $CO_2$ is reached in the *esm-1pctCO2* run is model-specific and depends on the carbon sink strengths of the individual land and ocean submodels. Assuming that about half of the emissions in the *1pctCO2* run remain in the atmosphere in most models [13], the timing of branching-off is likely reached after 60 to 70 years. We only analyze the evolution of atmospheric $CO_2$ in the output of fully coupled ESMs and do not consider EMICs (Earth systems model of intermediate complexity) in the ZecMIP ensemble, as they cannot represent certain pivotal processes. Please refer to [44] for detailed descriptions of the experiments and participating models.

## Model simulations of CMIP5 and CMIP6

We analyze climate-carbon simulations of ESMs participating in the fifth and sixth phase of the Coupled Model Intercomparison Project, *i. e.*, CMIP5 [47] and CMIP6 [52]. The model output is available from the Earth System Grid Federation, ESGF (https://esgf-data.dkrz.de/projects/esgf-dkrz/). Seven (eight) ESMs provide output for the variables of interest, *i. e.*, leaf area index (LAI) and net biome production (NBP), for simulations titled *RCP4.5*, *RCP8.5*, *1pctCO2* (*SSP2–4.5*, *SSP3–7.0*, *1pctCO2*) in CMIP5 (CMIP6). The models with the most complete representation of vegetation dynamics analyzed here are MPI-ESM-LR, MIROC-ESM, HadGEM2-ES in CMIP5 and MPI-ESM1.2-LR and UKESM1–0LL in CMIP6. The individual model setups and components are described in more detail in the specific Coupled Carbon Cycle Climate Model Intercomparison Project (C4MIP) for both model generations [17, 18].

In CMIP5, several Representative Concentration Pathways (RCPs) have been formulated describing different trajectories of greenhouse gas emissions, air pollutant production and land use changes for the 21[st] century. These scenarios have been designed based on projections of human population growth, technological advancement and societal responses [47, 55]. In CMIP6, these scenarios are updated with new estimates of emissions and land use projections produced with integrated assessment models and conflated with new future pathways of societal development, termed the Shared Socioeconomic Pathways (SSPs) [48, 52].

These RCP and SSP scenarios are $CO_2$ concentration-driven simulations, thus variations in carbon uptake of land and ocean have no effect on atmospheric $CO_2$ concentration. However, analyzing these simulations with different $CO_2$ growth rates across the different scenarios, but overall the same $CO_2$ forcing and initial conditions, allows us to determine if other ESMs also

exhibit timeline dependency. For both ensembles, we choose scenarios that follow a slow (*RCP8.5* and *SSP2–4.5*) and a medium-fast $CO_2$ growth rate (*RCP8.5* and *SSP3–7.0*). These simulations were initialized with the final state of the historical run in each ensemble and are simulated until the end of the 21st century. To obtain estimates for a rapid $CO_2$ growth rate, we also analyze output of the *1pctCO2* runs, initialized from a steady state of the pre-industrial control run and with atmospheric $CO_2$ concentration increasing by 1% each year until it quadruples the pre-industrial level.

As these sets of simulations also contain other forcing agents such as anthropogenic land-cover and land-use changes in mid-latitudes and tropical regions, we have to focus on the high-latitudes ($> 60$˚N) to distill the response of natural vegetation [9, 56]. The selection of latitudes exceeding 60˚N to exclude land-use changes is justified, given historical and projected extensive agricultural activities are located below this latitude [57]. We derive the changes in land carbon sink, calculated as the time integral of the NBP flux, and leaf area index (LAI), as proxy for vegetation changes, in simulations where $CO_2$ increases at different rates, but the overall increase is the same (for CMIP5: $\approx$ 155 ppm; for CMIP6: $\approx$ 200 ppm). To obtain comparability between CMIP5 and CMIP6, we report the results in form of sensitivities to 100 ppmv $CO_2$ change.

### Calculation and decompostion of TCRE

We derive and decompose TCRE primarily following the methodology detailed in Williams *et al.* [13]. Throughout this study, TCRE is computed using three methods: the expanding-window regression, delta, and a low-pass filter method. In the expanding-window regression method, TCRE is estimated using the slope in regressing the change in surface temperature against cumulative carbon emissions. The regression is computed iteratively in a expanding-window approach to monitor the temporal evolution and the stability of the TCRE estimate, excluding the initial ten years due to high natural variability. We report the average and standard deviation for the last ten regression estimates, *i.e.*, estimates close to full emission, and for all three realizations (Fig 1). The delta method calculates the change in surface warming relative to pre-industrial equilibrium, reporting mean and standard deviation over all three realization and final five years. For TCRE calculation, these values can be divided by the total carbon emissions of 1.2 Eg C, while plain temperature deltas are included in Table 1 for comparability with other estimates based on the 2 K carbon budget. In the low-pass filter method, we smooth year-to-year variability using a Savitzky-Golay filter and average the results across the three realizations for each experiment, as shown in Fig 3a.

Next, we describe the step-by-step decomposition of TCRE as detailed in Williams *et al.* [13]. This decomposition involves several equations, which reads as follows,

$$\text{TCRE} = \frac{\Delta T(t)}{I_{\text{E}}(t)} \tag{5}$$

$$= \left(\frac{\Delta T(t)}{\Delta C_{\text{A}}(t)}\right)\left(\frac{\Delta C_{\text{A}}(t)}{I_{\text{E}}(t)}\right) \tag{6}$$

$$= \left(\frac{\Delta T(t)}{\Delta F(t)}\right)\left(\frac{\Delta F(t)}{\Delta C_{\text{A}}(t)}\right)\left(\frac{\Delta C_{\text{A}}(t)}{I_{\text{E}}(t)}\right) \tag{7}$$

$$= \frac{1}{\lambda(t)}\left(1 - \frac{N(t)}{\Delta F(t)}\right)\left(\frac{\Delta F(t)}{\Delta C_{\text{A}}(t)}\right)\left(\frac{\Delta C_{\text{A}}(t)}{I_{\text{E}}(t)}\right). \tag{8}$$

The resulting equation describes the product of four terms: the inverse climate feedback parameter $\frac{1}{\lambda(t)}$, the radiative forcing fraction allocated to surface warming $\left(1 - \frac{N(t)}{\Delta F(t)}\right)$, the radiative forcing per unit atmospheric carbon $\frac{\Delta F(t)}{\Delta C_A(t)}$, and the airborne fraction $\frac{\Delta C_A(t)}{I_E(t)}$. The radiative forcing is estimated using the logarithmic dependence on atmospheric $CO_2$, *i.e.*,

$$\Delta F(t) = a \ln\left([CO_2](t)/[CO_2](t_0)\right), \tag{9}$$

where the radiative effective forcing coefficient $a$ is diagnosed from a simulation of atmospheric $CO_2$ quadrupling (abrupt-4xCO2). This coefficient is determined by utilizing the $y$-intercept of a regression fit for planetary heat uptake $N(t)$ versus surface warming $\Delta T(t)$ [13, 58]. To account for curvature in the relationship [13, 59], only the first twenty years of model output are used to calculate the fit. The ratios in Eq 8 are calculated using a low-pass filter (Savitzky-Golay filter) to mitigate interannual variability.

## Impulse-Response functions to predict atmospheric $CO_2$ in relaxation phase

Impulse-response functions are typically determined to describe the decrease in a substance (response, here atmospheric carbon) following a perturbation at time $t = 0$ (impulse, here carbon emission) [60]. Generally, an impulse-response function ($R$) describes a decay process that can be fitted by a sum of $n$ exponentials, *i. e.*, the superposition of multiple exponential functions, and reads

$$R(t) = \sum_{i=1}^{n} a_i \cdot \exp\left(\frac{-t}{\tau_i}\right) \text{ for } t \geq 0. \tag{10}$$

The value of $R$ at a given time $t$ is the fraction of the initial amount of a substance, here the emitted carbon, that is still airborne. Accordingly, the removed fraction $(1 - R)$ corresponds to the amount of emitted carbon absorbed in the ocean and on land. The parameter $a_i$ refers to the fraction of the emitted carbon that follows the decay associated with the $e$-folding time $\tau_i$, where $\sum_{i=1}^{n} a_i = 1$. The $e$-folding time refers to the time-scale in years for the value of $a_i$ to decrease to $\frac{1}{e}$ of its initial value. For example, if $a_1$ is 0.5 and $\tau_1$ is 100 years, then it takes 100 years for $a_1$ to decay to $\approx 0.185$ $(\frac{0.5}{e})$.

Accordingly, the equation

$$C(t) = C_T^0 \cdot R(t) = C_T^0 \cdot \sum_{i=1}^{n} \frac{C_i^0}{C_T^0} \cdot \exp\left(\frac{-t}{\tau_i}\right) \text{ for } t \geq 0 \tag{11}$$

describes the decay process of the total carbon that is airborne after emission cessation ($C_T^0$), where $\frac{C_i^0}{C_T^0}$ divides the total pool into $n$ sub-pools ($C_i^0$). For $n = 2$, the total airborne carbon is divided into a fast (F) and a slow (S) time-scale compartment, and we can rewrite Eq 11 as

$$C(t) = C_F^0 \cdot \exp\left(\frac{-t}{\tau_F}\right) + C_S^0 \cdot \exp\left(\frac{-t}{\tau_S}\right) \text{ for } t \geq 0 \tag{12}$$

where

$$C_T^0 = C_F^0 + C_S^0 \tag{13}$$

and $\tau_F$ and $\tau_S$ are the *e*-folding times for respective compartments. Combining Eqs 12 and 13 we obtain

$$C(t) = \eta C_T^0 \cdot \exp\left(\frac{-t}{\tau_F}\right) + ou(1 - \eta)C_T^0 \cdot \exp\left(\frac{-t}{\tau_S}\right) \text{ for } t \geq 0 \qquad (14)$$

where

$$\eta = \frac{C_F^0}{C_T^0}. \qquad (15)$$

Thus, the parameter $\eta$ refers to the fraction of the total carbon pool ($C_T^0$) that enters the fast decay compartment ($C_F^0$). We fit Eq 14 to the decay of atmospheric carbon as simulated in the different pathways and ZecMIP runs (Methods Section Idealized Pathway Experiments and Zero Emissions Commitment Model Intercomparison Project (ZecMIP) in CMIP6). Hence, the approach has only three fitting parameters, namely a scalar $\eta$ that divides the atmospheric carbon excess into a fast and a slow pool, and the respective parameters $\tau_F$ and $\tau_S$ that determine the decay rates (*e*-folding times) of each pool. In a second fitting stage, we sample $\eta$ from the parameter space determined when $\eta$ is also a fitting parameter, as listed in S1 Table, to obtain more reliable uncertainty estimates for the parameters $\tau_F$ and $\tau_S$ (S2 Table). We use the Python nonlinear curve fitting package LMFIT [61] to optimize the three parameters.

## Supporting information

**S1 Fig. Change in atmospheric CO$_2$ concentration across the different pathways and different periods for the same amount of emitted carbon. a–d** Relationship between cumulative carbon emissions (*x*-axis) and the change in atmospheric CO$_2$ concentration ($\Delta$[CO$_2$]) is shown, where the colors refer to the four different pathways analogous to Fig 1 and the markers refer to the different time periods. The arrows annotate the final $\Delta$[CO$_2$] in each model experiment. The dotted black line facilitates comparison between the pathways.
(TIF)

**S2 Fig. Pathway dependence of the global carbon cycle response translates into different warming potential of the land for a given emission.** Transient Climate Response to Cumulative Emissions (TCRE) for different pathways (*x*-axis) and simulation periods (colors), focusing on the warming of the land surface. Land TCRE is estimated using the conventionally used linear regression method [13]. Shaded dots exhibit the spread in the estimates of the final five years when 1.2 Eg C have been emitted as well as among different realizations. Pointplot with whiskers show the mean and standard deviation of the spread.
(TIF)

**S3 Fig. Vegetation dynamics explain the pathway dependence in the terrestrial carbon cycle. a** Global map of the difference in the total forest carbon pool between the L and E pathways (yellow and red in b and c, respectively) at the end of the emission phase, calculated with MPI-ESM1–2-LR in the 200-year runs. **b** Change in carbon pool for forest, grass- and shrublands, and total vegetation for the different emission pathways. The whiskers represent the uncertainty between the different realizations. **c** Change of forest carbon pool across the four pathways as a function of time for both the emission and relaxation phases.
(TIF)

**S4 Fig. Evolution of global mean surface air temperature across pathways after emission stop. a** Different colored dots depict the different emission pathways smoothed by a low-pass

filter (Savitzky-Golay filter), where the lines represent a nonparametric LOWESS fit. **b** as in **a**, but with atmospheric $CO_2$ as *x*-axis.
(TIF)

**S5 Fig. Leaf area sensitivity to different $CO_2$ growth rates in CMIP ensembles. a** Change in leaf area index at latitudes above 60˚ N per a 100 ppm change in atmospheric $CO_2$ across three different future scenarios in CMIP6 reflecting different $CO_2$ growth rates (*x*-axis). The black dot refers to the multi-model mean value for each growth rate, where the whiskers represent the standard deviation. **b** as in c but for CMIP5.
(TIF)

**S6 Fig. Sink efficiency differs across pathways after emission cessation of the same amount of carbon.** Sink efficiency for different pathways (*x*-axis) and simulation periods (colors) at the end of the emission phase. Shaded dots exhibit the spread in the estimates of the final five years when 1.2 Eg C have been emitted as well as among different realizations. Pointplot with whiskers show the mean and standard deviation of the spread.
(TIF)

**S7 Fig. Partitioning of emissions into land and oceans sinks across the pathways in the EMIC CLIMBER-2. a** Cumulative land and ocean sinks are juxtaposed across all pathways. The markers denote the final state at the end of the emission phase of 1.2 Eg C. The dashed line indicates equal magnitude of cumulative land and ocean sinks. The gray and black lines refer to the relationships inferred from MPI-ESM1.2-LR simulations and GCP estimates, respectively. **b** As in a, but for relaxation phase. The Climate-Biosphere model (version 2) CLIMBER-2 is an Earth System Model of Intermediate Complexity (EMIC) and consists of a statistical-dynamical atmosphere component (51˚ × 10˚ spatial resolution), a 2D ocean component with three zonally averaged basins, and a land component including dynamic vegetation [49]. The same model version of CLIMBER-2 is used, which was also used for the ZecMIP runs [6].
(TIF)

**S8 Fig. Pronounced pathway dependence in characteristic entities of the Earth system as simulated by the MPI-ESM1–2-LR. a**. Thermosteric sea-level change for given emission of 1 Eg C for different pathways estimated using a linear regression method analogous to TCRE in Fig 1d; [13]. Shaded dots exhibit the spread in the estimates of the final five years when 1.2 Eg C have been emitted as well as among different realizations. Pointplot with whiskers show the mean and standard deviation of the spread. **b** as in **a** but for surface-ocean pH change. **c** as in **a** but for sea-ice volume change.
(TIF)

**S1 Table. Best-fit values for parameters of the double exponential decay function (see Eq 14).**
(PDF)

**S2 Table. Best-fit values for parameters of the double exponential decay function (Eq 14), where $\eta$ is sampled from the parameter space determined in S1 Table.**
(PDF)

## Acknowledgments

AJW thanks Ralph Keeling, Guilherme Torres Mendonça, and Matteo Puglini for stimulating discussions on the topic.

## Author Contributions

**Conceptualization:** Alexander J. Winkler, Ranga Myneni, Christian Reimers, Markus Reichstein, Victor Brovkin.

**Data curation:** Alexander J. Winkler.

**Formal analysis:** Alexander J. Winkler.

**Funding acquisition:** Markus Reichstein, Victor Brovkin.

**Investigation:** Alexander J. Winkler.

**Methodology:** Alexander J. Winkler, Christian Reimers.

**Project administration:** Alexander J. Winkler.

**Software:** Alexander J. Winkler.

**Validation:** Alexander J. Winkler.

**Visualization:** Alexander J. Winkler.

**Writing – original draft:** Alexander J. Winkler, Ranga Myneni.

**Writing – review & editing:** Alexander J. Winkler, Ranga Myneni, Christian Reimers, Markus Reichstein, Victor Brovkin.

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
