## [Decision Letter · Decision Letter 0]

7 Dec 2023

PONE-D-23-32708Carbon System State Determines Warming Potential of EmissionsPLOS ONE

Dear Dr. Winkler,

Thank you for submitting your manuscript to PLOS ONE. After careful consideration, we feel that it has merit but does not fully meet PLOS ONE’s publication criteria as it currently stands. Therefore, we invite you to submit a revised version of the manuscript that addresses the points raised during the review process.

We look forward to receiving your revised manuscript.

Kind regards,

Yangyang Xu

Academic Editor

PLOS ONE

6. Thank you for stating the following financial disclosure: 

 [VB and AJW acknowledge the Deutsche Forschungsgemeinschaft (DFG, German Research Foundation) under Germany‘s Excellence Strategy – EXC 2037 ’CLICCS - Climate, Climatic Change, and Society’ – Project Number: 390683824".

MR, CR and AJW acknowledge support by the European Research Council (ERC) Synergy Grant "Understanding and Modelling the Earth System with Machine Learning (USMILE)" under the Horizon 2020 research and innovation 415 programme (Grant agreement No. 855187). 

RBM and AJW acknowledge funding by the Alexander von Humboldt Foundation. 

RBM also acknowledges funding from NASA Earth Science Divison].  

7. Thank you for stating the following in the Acknowledgments Section of your manuscript: 

[AJW thanks Ralph Keeling, Guilherme Torres Mendonça, and Matteo Puglini

for stimulating discussions on the topic. The study was funded by the Deutsche

Forschungsgemeinschaft (DFG, German Research Foundation) under Germany‘s

Excellence Strategy – EXC 2037 ’CLICCS - Climate, Climatic Change, and Society’

– Project Number: 390683824". MR, CR and AJW acknowledge support by the Eu261 ropean Research Council (ERC) Synergy Grant "Understanding and Modelling

the Earth System with Machine Learning (USMILE)" under the Horizon 2020

research and innovation 415 programme (Grant agreement No. 855187). RBM

and AJW acknowledge funding by the Alexander von Humboldt Foundation.

RBM also acknowledges funding from NASA Earth Science Divison.]

 [VB and AJW acknowledge the Deutsche Forschungsgemeinschaft (DFG, German Research Foundation) under Germany‘s Excellence Strategy – EXC 2037 ’CLICCS - Climate, Climatic Change, and Society’ – Project Number: 390683824".

MR, CR and AJW acknowledge support by the European Research Council (ERC) Synergy Grant "Understanding and Modelling the Earth System with Machine Learning (USMILE)" under the Horizon 2020 research and innovation 415 programme (Grant agreement No. 855187). 

RBM and AJW acknowledge funding by the Alexander von Humboldt Foundation. 

RBM also acknowledges funding from NASA Earth Science Divison].

8. We note that you have indicated that data from this study are available upon request. PLOS only allows data to be available upon request if there are legal or ethical restrictions on sharing data publicly. For more information on unacceptable data access restrictions, please see http://journals.plos.org/plosone/s/data-availability#loc-unacceptable-data-access-restrictions. 

9. We note that you have stated that you will provide repository information for your data at acceptance. Should your manuscript be accepted for publication, we will hold it until you provide the relevant accession numbers or DOIs necessary to access your data. If you wish to make changes to your Data Availability statement, please describe these changes in your cover letter and we will update your Data Availability statement to reflect the information you provide.

10. Please amend your manuscript to include your abstract after the title page.

11. We note that Figure 2A and 4A in your submission contain [map/satellite] images which may be copyrighted. All PLOS content is published under the Creative Commons Attribution License (CC BY 4.0), which means that the manuscript, images, and Supporting Information files will be freely available online, and any third party is permitted to access, download, copy, distribute, and use these materials in any way, even commercially, with proper attribution. For these reasons, we cannot publish previously copyrighted maps or satellite images created using proprietary data, such as Google software (Google Maps, Street View, and Earth). For more information, see our copyright guidelines: http://journals.plos.org/plosone/s/licenses-and-copyright.

a. You may seek permission from the original copyright holder of Figure 2A and 4A to publish the content specifically under the CC BY 4.0 license.  

12. Please include captions for your Supporting Information files at the end of your manuscript, and update any in-text citations to match accordingly. Please see our Supporting Information guidelines for more information: http://journals.plos.org/plosone/s/supporting-information. 

Reviewers' comments:

Reviewer's Responses to Questions

**Comments to the Author**

1. Is the manuscript technically sound, and do the data support the conclusions?

Reviewer #1: Yes

Reviewer #2: Partly

2. Has the statistical analysis been performed appropriately and rigorously? 

Reviewer #1: I Don't Know

Reviewer #2: Yes

3. Have the authors made all data underlying the findings in their manuscript fully available?

Reviewer #1: Yes

Reviewer #2: No

4. Is the manuscript presented in an intelligible fashion and written in standard English?

Reviewer #1: Yes

Reviewer #2: Yes

5. Review Comments to the Author

Reviewer #1: Thanks for writing this paper. It is a relevant and overlooked issue, and something that has worried me for some time. It is nice to see an analysis of this. As you show, the path dependence is much stronger for the carbon response, compared to the temperature response (TCRE). The way you frame the paper, it feels like there is a big path dependence on TCRE. There is a counteracting path dependency in the forcing to temperature response as well. This is something that could be explored or brought out a little more.

Overall, I don’t have any major problems with the paper, but I have lots of clarifications, minor comments, and requests.

1. Abstract: “total anthropogenic carbon emissions”. To make sure there is no ambiguity, probably makes sense to put cumulative in there. “total anthropogenic cumulative carbon emissions”, or similar

2. The text in Supporting Information Text S1 is good, and if possible, I would try and integrate it into the main text. You mention many of these issues in the main text, so it is a matter of just layering in any differences.

3. Table 1: Is 5 years averaging long enough to get the interannual variability out?

4. Figure 1c, shows the [CO2] versus cumulative CO2. It would be worth putting in the equivalent figure for T versus cumulative CO2. I see you have this in Figure S2. Figure S2 shows a much stronger linearity, and it is this linearity that is referred to in the TCRE relationship. There is still a state dependence, but it is not as big. However, there are really two state dependencies here, which are playing out in opposite directions: the [CO2] versus cumulative CO2 and the temperature versus [CO2]. Presumably, the state dependence of temperature versus [CO2] is completely opposite to the equivalent for [CO2]? Is it possible to somehow shows this. I do not know the best way to plot this, but at each year is it possible to calculate something like T/[CO2] versus cumulative CO2? Not sure it makes sense. There is the style of figure in Matthews et al 2009, but that is with time and not cumulative emissions. Perhaps this would mean an extra figure with subpanels in the paper, but it would be really nice to have a set of figures showing the CO2 sensitivity, the temperature sensitivity, and how they offset each other to produce Figure S2. I think this would greatly strengthen the paper. Though, it shows that the state dependence is less at the aggregate, important at the more disaggregated level.

5. I did not get how you calculated the TCRE, you write “TCRE is estimated using the conventionally used linear regression method7”. But they seem to take the last 20 years, and I don’t see they did a regression. A regression of the entire timeseries will give a different number, compared to the last 5-20 years. I think it is more conventional to get the TCRE using a 1% CO2 experiment, and the temperature when emissions go to zero. Just clarify how you did this, and note if it is important, as the slope may not be the same as last 5-20 years average temperature divided by cumulative emissions. Does the regression goes through zero?

6. Figure 1a. It was a little bit ambiguous on the 100 and 200 years. You either distribute the emissions over 100 years, or over 200 years? Basically, the 100 years version does everything at twice the rate? Perhaps try and make that more clear.

7. Figure 2. The vegetation dynamics. These simulations just use preindustrial vegetation? I assume there is no LUC? So the difference in carbon in forests, etc, is purely due to feedbacks on the land system? If you have age dynamics, it will be important how the system is initialised? Also, you do this >60N, to take out LUC, is that a robust choice?

8. Figure 3, panel a and b. Again, showing the temperature equivalent of this would be useful. This would be ZEC? Basically, the ZEC is path dependent (already clear in ZECMIP). But the temperature response must somehow offset the variation in the carbon response? (also in ZECMIP)

9. Figure 3, panel c and d. I did not get how you constructed the figure. You did an additional simulation? How could you constrain CMIP6 to 2.2ppm/yr, etc? Or was this different SSPs? If so, write it down.

10. Sink efficiency. Line 196. Is this defined the same as Raupach et al, equation 14. Can you write this as an equation to make it less ambiguous?

11. The slow and fast time scales, are based on the Figures 3a and b? And this is after zero emissions? After zero, the models generally only run to 100 years? How then can you have confidence in fitting time scales of 750 years or so? There is very little knowledge to fit that time scale. I imagine that you could force 200 years as the long time scale, and maybe get a similar fit?

12. Lines 15-17: Though, this is really referring to Figure S2, where the path dependence seems much weaker?

13. Line 57: “Business as usual scenario for 2C” What is that? Perhaps just say a 2C scenario.

14. Lines 70-71: If forcing is something like log(CO2), then as emissions and concentrations decline, the forcing will also decline. So that declining concentration phase, must be offset by the forcing change? Basically, Figure 1c and Figure S2 have to balance somehow.

15. Lines 77+. You say you get the TCRE by a regression. So would you put a linear fit through the green circles in Figure S2c? Or do you get the last five years temperature? If anything, Figure S2 shows how really important it becomes on how you estimate TCRE. I would think this is a point that really needs to be highlighted.

16. Line 128+. You are showing that the ZEC is path dependent (yes, you see this in MacDougall et al)

17. Lines 143+: I have really low confidence in these long time scales here, given that they are so far outside the simulation domain.

18. Lines 222-224: Well, I think Figure S2 sort of shows the opposite. Despite the different pathways, they all generally show that T is proportional to cumulative CO2, the difference is sort of in the second digit?

19. Lines 225: “It is not the amount emitted…” Well, CO2 = cumulative emissions - cumulative land sink - cumulative ocean sink. I guess I could plot temperature as a function of either of the three cumulative terms (Figure 4, Figure S7). It would be worth doing Figure S7 for each of the cumulative terms, which one is more robust description of T? I am not sure why in Figure 4 you make the choices on which you will use for the x-axis. Presumably, Figure 4c and d would be similar with cumulative emissions?

20. Figure 1a. Can you give equations for the curves, in the methods or something.

21. One implication of your paper is that emission driven runs are important. This preprint just out is relevant in this context. https://egusphere.copernicus.org/preprints/2023/egusphere-2023-2127/

22. Particularly for your ZEC analysis and fast and slow time scales, Jenkins et al is a relevant paper. It is probably worth having a look and making the relevant connections if they exist. https://agupubs.onlinelibrary.wiley.com/doi/full/10.1029/2022GL101047

23. This paper may also be relevant for the ZEC: https://iopscience.iop.org/article/10.1088/1748-9326/acab1a

24. Overall, you are showing that TCRE and ZEC are path dependent, and that is an important point to make, and not sufficiently understood.

Reviewer #2: See attached pdf for all comments. The rest of this comment is filler to reach the minimum character count.

Filler Filler Filler Filler Filler Filler Filler Filler Filler Filler Filler Filler Filler

6. PLOS authors have the option to publish the peer review history of their article (what does this mean?). If published, this will include your full peer review and any attached files.

Reviewer #1: No

Reviewer #2: No

---

## [Author Response · Author response to Decision Letter 0]

3 Apr 2024

We uploaded two documents containing detailed point-by-point responses to the reviewer comments.

---

## [Decision Letter · Decision Letter 1]

30 Apr 2024

PONE-D-23-32708R1Carbon System State Determines Warming Potential of Emissions

PLOS ONE

Dear Dr. Winkler,

Thank you for submitting your manuscript to PLOS ONE. After careful consideration, we feel that it has merit but does not fully meet PLOS ONE’s publication criteria as it currently stands. Therefore, we invite you to submit a revised version of the manuscript that addresses the points raised during the review process.

In addition to R1's comments, please also consider and respond to the following comments sent to me directly from R2 during the 2nd round of review.

For what it's worth, I still think the authors need to do a better job of explaining how their conclusion fits with current knowledge. I don't think a paper based on a single model that says, "Throw away everything you thought you knew about TCRE being pathway independent", without explaining exactly how they come to such a strong conclusion, is justifiable. If they can say something like, "The TCRE appears pathway independent because all our simulations basically use the same sort of carbon cycle state", (or some other explanation of why it has appeared that the TCRE is pathway-independent for so long and no-one has noticed that it isn't), that would be great. Alternately, they might just say, "All the effects we're looking at here lead to deviations of ~10% from a pathway-independent TCRE", in which case their work is completely in line with previous work e.g. https://esd.copernicus.org/preprints/esd-2023-7/ and https://iopscience.iop.org/article/10.1088/1748-9326/ab83af 

We look forward to receiving your revised manuscript.

Kind regards,

Yangyang Xu

Academic Editor

PLOS ONE

Journal Requirements:

Additional Editor Comments:

In addition to Reviewer #1' comments, please also consider the following comments sent to me directly from Reviewer #2:

For what it's worth, I still think the authors need to do a better job of explaining how their conclusion fits with current knowledge. I don't think a paper based on a single model that says, "Throw away everything you thought you knew about TCRE being pathway independent", without explaining exactly how they come to such a strong conclusion, is justifiable. If they can say something like, "The TCRE appears pathway independent because all our simulations basically use the same sort of carbon cycle state", (or some other explanation of why it has appeared that the TCRE is pathway-independent for so long and no-one has noticed that it isn't), that would be great. Alternately, they might just say, "All the effects we're looking at here lead to deviations of ~10% from a pathway-independent TCRE", in which case their work is completely in line with previous work e.g. https://esd.copernicus.org/preprints/esd-2023-7/ and https://iopscience.iop.org/article/10.1088/1748-9326/ab83af

Reviewers' comments:

Reviewer's Responses to Questions

**Comments to the Author**

1. If the authors have adequately addressed your comments raised in a previous round of review and you feel that this manuscript is now acceptable for publication, you may indicate that here to bypass the “Comments to the Author” section, enter your conflict of interest statement in the “Confidential to Editor” section, and submit your "Accept" recommendation.

Reviewer #1: All comments have been addressed

2. Is the manuscript technically sound, and do the data support the conclusions?

Reviewer #1: Yes

3. Has the statistical analysis been performed appropriately and rigorously? 

Reviewer #1: Yes

4. Have the authors made all data underlying the findings in their manuscript fully available?

Reviewer #1: Yes

5. Is the manuscript presented in an intelligible fashion and written in standard English?

Reviewer #1: Yes

6. Review Comments to the Author

Reviewer #1: I thank the authors for a very comprehensive response to comments! I think the article is ready to see the light of day, and I am happy to accept.

I do have a few minor comments, the authors may wish to take up. I do not need to review again.

• Fig 1d. I would prefer if this showed the delta T versus cumulative CO2 figure here, like 1c, but for T. The current 1d can go in the SI. You discuss Figure Sup 2 so much, and it would just be very useful to have the full TCRE figure here

• Fig 2b. I would write “Cumulative airborne fraction” to avoid confusion

• Fig 2. I see how you tried to use the arrows to show the panel connections, though, it took a while. You could write in each panel things like TCRE = b*c, etc. Or you could, maybe, do a 3x3 panel: TCRE, TCR, CAF in the first, TCR (yes repeat), d*e, etc. So if you can figure something out, as it was useful. Maybe even you could put the equations in the bottom right, in the white space? (Just ideas, you decide what works)

• Fig Sup S2. It could be useful, even if a little messy, to add the regression lines for each curve used to calculate the TCRE. I still feel uneasy with these regression fits, when the data may not be quite linear. It would be good to visualise if it was a problem or not…

7. PLOS authors have the option to publish the peer review history of their article (what does this mean?). If published, this will include your full peer review and any attached files.

Reviewer #1: No

---

## [Author Response · Author response to Decision Letter 1]

4 Jun 2024

Please see attached files displaying point-by-point responses to the reviewers' comments.

---

## [Decision Letter · Decision Letter 2]

12 Jun 2024

Carbon System State Determines Warming Potential of Emissions

PONE-D-23-32708R2

Dear Dr. Winkler,

We’re pleased to inform you that your manuscript has been judged scientifically suitable for publication and will be formally accepted for publication once it meets all outstanding technical requirements.

Kind regards,

Yangyang Xu

Academic Editor

PLOS ONE

Additional Editor Comments (optional):

Reviewers' comments:

Reviewer's Responses to Questions

**Comments to the Author**

1. If the authors have adequately addressed your comments raised in a previous round of review and you feel that this manuscript is now acceptable for publication, you may indicate that here to bypass the “Comments to the Author” section, enter your conflict of interest statement in the “Confidential to Editor” section, and submit your "Accept" recommendation.

Reviewer #1: All comments have been addressed

2. Is the manuscript technically sound, and do the data support the conclusions?

Reviewer #1: Yes

3. Has the statistical analysis been performed appropriately and rigorously? 

Reviewer #1: Yes

4. Have the authors made all data underlying the findings in their manuscript fully available?

Reviewer #1: Yes

5. Is the manuscript presented in an intelligible fashion and written in standard English?

Reviewer #1: Yes

6. Review Comments to the Author

Reviewer #1: (No Response)

7. PLOS authors have the option to publish the peer review history of their article (what does this mean?). If published, this will include your full peer review and any attached files.

Reviewer #1: No

---

## [Editor Report · Acceptance letter]

25 Jun 2024

PONE-D-23-32708R2 

PLOS ONE

Dear Dr. Winkler, 

I'm pleased to inform you that your manuscript has been deemed suitable for publication in PLOS ONE. Congratulations! Your manuscript is now being handed over to our production team.

Kind regards, 

on behalf of

Dr. Yangyang Xu 

Academic Editor

PLOS ONE